# Synergistic Effects of Regular Walking and Alkaline Electrolyzed Water on Decreasing Inflammation and Oxidative Stress, and Increasing Quality of Life in Individuals with Type 2 Diabetes: A Community Based Randomized Controlled Trial

**DOI:** 10.3390/antiox9100946

**Published:** 2020-10-01

**Authors:** Yohanes Andy Rias, Adi Lukas Kurniawan, Ching Wen Chang, Christopher James Gordon, Hsiu Ting Tsai

**Affiliations:** 1School of Nursing, College of Nursing, Taipei Medical University, Taipei 11031, Taiwan; yohanes.andi@iik.ac.id; 2Faculty of Health and Medicine, College of Nursing, Institut Ilmu Kesehatan Bhakti Wiyata, Kediri 64114, Indonesia; 3Research Center for Healthcare Industry Innovation, National Taipei University of Nursing and Health Sciences, 365 Ming-te Road, Beitou District, Taipei 112, Taiwan; 8lukas@ntunhs.edu.tw; 4Department of Obstetrics and Gynecology, Taipei Medical University Hospital, Taipei 11031, Taiwan; cwchang967@tmu.edu.tw; 5Susan Wakil School of Nursing and Midwifery, Faculty of Medicine and Health, The University of Sydney, Camperdown 2050, Australia; christopher.gordon@sydney.edu.au; 6Post-Baccalaureate Program in Nursing, College of Nursing, Taipei Medical University, Taipei 11031, Taiwan

**Keywords:** alkaline electrolyzed water, inflammation marker, oxidative stress, regular walking, synergistic effects, type 2 diabetes, quality of life

## Abstract

Alkaline electrolyzed water (AEW) and walking are strongly recommended for ameliorating oxidative stress and inflammation. Nevertheless, there is a lack of information on the combination of both on alleviating inflammation, oxidative stress, and improving the quality of life (QoL). We investigated the synergistic effects of drinking AEW and walking on advanced glycation end products (AGEs), advanced oxidation protein products (AOPPs), malondialdehyde (MDA), white blood cells (WBCs), neutrophil-lymphocyte ratio (NLR) and QoL. In total, 81 eligible patients with type 2 diabetes (T2DM) were randomly allocated via single blind to four groups: consumed 2 L/day of AEW (*n* = 20), instructed to walk for 150 min/week (*n* = 20), received a combination of AEW and walking (*n* = 20), and continue their habitual diet and activity (*n* = 21). Data were collected and analyzed before and after 8 weeks of intervention. Our results showed a significant interaction between the group and time, with both AEW and walking independently and synergistically ameliorating AGEs, AOPPs, MDA, NLR and WBCs levels. Moreover, the AEW group had a higher physical and total QoL score. The walking group and the combined group had higher scores in physical, mental and total QoL compared to the control group. The synergistic effect of AEW and regular walking are an advisable treatment for patients with T2DM.

## 1. Introduction

Type 2 diabetes (T2DM) is characterized by insulin resistance, wherein there is an overproduction and secretion of insulin in the early stage which lead to reduced insulin secretion and pancreatic beta cell death at late states. It has been reported that oxidative stress and inflammation are the risk factors for developing T2DM, and major causes of the complications and mortality among patients with T2DM [1,2,3]. The International Diabetes Federation has projected that the number of people with diabetes worldwide will reach 578 million by 2030, and will rise to 700 million by 2045. In particular, this threat is also present in Indonesia, where the prevalence of diabetes is expected to rise from 10.7 million to 16.6 million from 2019 and 2045 [4]. Consequently, these conditions may impair the quality of life (QoL) [5]. Therefore, identifying the formation of oxidative stress and inflammatory markers, and their roles in behavioral factors, is necessary for a better understanding of their effects on the QoL.

Oxidative stress biomarkers, such as advanced glycation end products (AGEs), advanced oxidation protein products (AOPPs) and malondialdehyde (MDA), are crucial parameters that facilitate more-individualized treatments of T2DM. Furthermore, higher levels of these markers are powerful predictors for developing vascular complications [6], and higher levels of inflammatory biomarkers such as white blood cells (WBCs) and the neutrophil-lymphocyte ratio (NLR) were detected in T2DM patients with inflammatory abnormalities [7,8]. Researchers concluded that both WBCs and the NLR are key pathogenetic mechanisms triggering T2DM, and detecting the NLR is relatively inexpensive [7]. Moreover, a study in 2013 reported that an NLR of > 2.89 was considered a predictor of vascular complications for T2DM in diabetic geriatric patients [9]. This indicated that detecting predictive biomarkers of vascular complications in T2DM, such as AGEs, AOPPs, MDA, the NLR and WBCs, is therefore necessary. Moreover, it is important to note that oxidative stress and inflammatory markers have been integrated and developed into research protocols to provide health professionals a way to recognize underlying biological processes and assess precision interventions [10]. Notably, a biobehavioral approach is one of the greatest challenges, and it is important to integrate biological phenomena with psychosocial and behavioral changes in health-related outcomes such as QoL. Further, links between biological markers and behavior in relation to health outcomes can help form basic strategies, such as physical activity and nutritional therapy, to restore, maintain and promote health [10].

It was found that regular aerobic exercise effectively reduces fasting blood glucose (FBG) and AOPPs and maintains a stable blood pressure [11]. Regular physical activity was reported to improve QoL and prevent diabetic complications through attenuating the formation of AGEs in patients with T2DM [12]. Furthermore, an animal study demonstrated that low-intensity exercise reduced the MDA level in rats with streptozotocin-induced diabetes [13]. Furthermore, regular exercise three or more times per week was inversely associated with a lower NLR [14], and there was a large decrease in WBCs [15]. However, approximately 46% of the general population of Indonesia are physically inactive [16]. Behavioral interventions such as regular walking at least 30 min/day can serve as convenient and easy ways to provide interventions for patients with T2DM [17]. However, no study has investigated the effects of regular walking on reducing levels of AGEs, AOPPs, MDA, the NLR, or WBCs in individuals with T2DM, particularly for Indonesians. Hence, examining the associations of walking with an increased QoL and the prevention of diabetic complications among Indonesians with T2DM is important.

Alkaline electrolyzed water (AEW) is produced near the cathode during water electrolysis. AEW is a common health-beneficial drink that acts as a medical device that exhibits the alkaline power of hydrogen (pH 8~10), and has negative oxidation-reduction potential (ORP), which has the ability to scavenge reactive oxygen species (ROS) and reduce inflammation, and can protect deoxyribonucleic acid from oxidative damage [18]. AEW is also used as a therapeutic substance, and was reported to reduce insulin resistance (IR) in animal models [19,20]. However, few studies have explored the antidiabetic effects of AEW in humans [21,22,23,24,25,26]. AEW was revealed to ameliorate blood glucose, MDA, interleukin-6 and tumor necrosis factor alpha [25], and to increase the oxidant potential [26]. A study by Gadek et al. [27] showed that patients who drank at least 2 L/day of AEW had significantly decreased blood glucose and glycated hemoglobin levels after treatment for 6 days. A multicenter prospective double-blind randomized control trial suggested that AEW significantly decreased IR and maintained oxidative stress, as assessed by measurements of biological antioxidant potential levels [21]. Further, a previous study showed that treatment with 2 L/day of AEW for patients receiving radiotherapy for liver tumors improved QoL within 6 weeks [28]. Moreover, lower levels of inflammation markers, such as NLR and WBCs, were positively associated with higher QoL [5]. Similarly, the amelioration of chronic inflammation is a potentially novel biobehavioral therapeutic strategy for improving QoL [10]. However, the effects of AEW on AGEs, AOPPs, MDA, the NLR, WBCs and the QoL have not been investigated in patients with T2DM. Interestingly, it was found that AEW has beneficial effects on exercise due to its free radical-scavenging properties, thereby enabling the maintenance of muscle performance and redox homeostasis during consecutive days of exercise [29]. This highlights the fact that regular walking together with AEW may have positive effects on reducing inflammation and oxidative stress, which consequently increase QoL in individuals with T2DM. However, these relationships require clarification.

Therefore, we conducted a randomized control trial study to investigate the effects of regular walking, the consumption of AEW, and their combined synergistic effect on reducing levels of AGEs, AOPPs, MDA, the NLR and WBCs, and on increasing the QoL of patients with T2DM. The objectives of this study were to assess the effects of regular walking, drinking AEW, and their combined synergistic effect on ameliorating oxidative stress (AGEs, AOPPs, and MDA) and inflammatory markers (the NLR and WBCs), FBG, and on promoting the QoL among Indonesians with T2DM, over a period of 8 weeks.

## 2. Materials and Methods

### 2.1. Research Ethic and Sample Size

The study protocol was reviewed and approved by the Clinical Research Ethics Committee of Siti Khotidjah Muhammadiyah Sepanjang Hospital (CRE; 009/KET-TPEP/X-2018) and Institutional Review Board Ethics Committee of Institute Health Science Strada Indonesia (IRB; 326/KEPK/II/2018) and conformed to the provisions of the *Declaration of Helsinki*. Written informed consent was obtained from each participant after they had received both verbal and written information about the research. Moreover, the trial study was registered with the Thai Clinical Trials Registry (Identifier: TCTR20200403002). The data was extracted electronically in a comma-separated variable format for a statistical software package. Databases were maintained on a secured server within the password-protected computer system and data were anonymized. Only members of this research study team will have access to the data. The electronic data will be deleted within five years following publication. Sample size calculation in this study used G-Power statistical software based on prior research [30]. Our study estimated the sample size based on an *F*-test, an a priori type of power analysis, with an effect size of 0.25, a power (1–β) of 0.8, and an α level of 0.05, for four groups (control group, regular walking group, AEW group, and AEW and regular walking combined group). This method produced a total estimated sample size of 76. In this study, the sample size of each group was 19 subjects. We estimated a 5% dropout rate and decided to increase the sample size to 20 subjects in each group.

### 2.2. Study Design and Setting

A randomized control trial with a four parallel intervention group design was conducted to evaluate the effects of regular walking, AEW supplementation, and the two combined on ameliorating oxidative stress and inflammatory markers, and improving the QoL among Indonesians with T2DM, with 8 weeks intervention. Individuals with T2DM were recruited from two community clinics between 1 July and 30 November, 2018 in East Java, Indonesia. In this study, there were 154 patients with T2DM identified from medical records at recruitment; 81 met the study inclusion and exclusion criteria. The inclusion criteria were (1) Indonesian nationals aged 17~80 years, (2) those who agreed to participate in the study, (3) patients who were diagnosed with T2DM at least six months before, (4) no previous myocardial infarction, stroke or diagnosed coronary artery disease, (5) those who visited the community clinic outpatient department and completed the questionnaires, (6) those who were diagnosed with T2DM as confirmed by a physician with 2 h blood glucose test results of >200 mg/dL or FBG of >126 mg/dL [31], and (7) all patients were on stable oral hypoglycemic agents (Metformin and/or Glibenclamide). The exclusion criteria included participants who (1) had experience of drinking AEW, (2) had a Mini-Mental State Exam score of ≤24, (3) had an amputated limb or could not walk, (4) were pregnant, (5) used antidepressants, (6) had auditory deficiencies, (7) had previous thrombotic events, cancer, autoimmune disease or other chronic and acute disease, (8) patients who were on insulin injections, and (9) those who had physical and mental problems that prevent walking. All the medication were distributed by physicians and no changes were allowed. Recruited patients with T2DM were randomly allocated to the four experimental groups. The CONSORT diagram of participant recruitment and flow is presented in Figure 1.

### 2.3. Randomization and Masking

The group assignment sequence was generated by a computer (www.randomizer.org) with a ratio of 1:1, and the sequence was performed by research assistants who were not involved in data collection. Once a patient agreed to participate in the study, this information was passed by the research assistant to a clinical nurse who then allocated the patients to the groups based on an opaque and sealed envelope with the identifiers that were given to all participants according to their sequence of entering the trial. It was impossible to blind the participants due to the nature of the interventions.

### 2.4. Interventions

#### 2.4.1. Alkaline Electrolyzed Water Group

As recommended by other studies [22,24,27], participants in the alkaline electrolyzed water group were asked to drink 2 L/day of alkaline electrolyzed water at pH 9.5 and with an ORP of −400.1 ± 13.3 mV for 8 weeks. AEW was produced by a Kangen Water Type SD501 Platinum Machine (Tokyo, Japan; ISO 9001:2015 and 14001:2015 certified) and was packaged in a dark bottle to maintain the stability of hydrogen; the alkaline electrolyzed water was prepared fresh twice a week. We would like to confirm that all patients collected the AEW twice a week. Moreover, AEW ingested during the experiment consisted of 2.47 mg/L of chloride (CI^−^), copper (Cu; 0.147 mg/L), iron (Fe; 0.121 mg/L), nitrate (NO3^−^; 0.093 mg/L), nitride (NO2^−^; 0.022 mg/L), calcium carbonate (CaCO_3_; 62.33 mg/L), aluminum (AI; 0.05 mg/L) and sulfate (SO4^2−^; 0.5 mg/L), produced in a private laboratory (ISO 14001: 2015 and 9001:2015 certified), and was approved as a water consumable. Research assistants controlled the information report 3 days per week and encouraged participant adherence to the intervention protocol for safety management. Data were collected at the baseline before alkaline electrolyzed water intervention (T0) and post-intervention at 8 weeks (T1) to investigate the effects of the alkaline electrolyzed water interventions.

#### 2.4.2. Walking Group

Participants in the regular walking group were instructed to walk around the house, park or roads for a duration at least 30 min, five times/week (for a total of 150 min/week) for 8 weeks, without measuring the distance or speed [32]. Self-reported logbooks were provided for participants to record the frequency and duration of their regular walks. Research assistants controlled the information report 3 days per week and encouraged participants to adhere to the intervention protocol for safety management. Data were collected at the baseline before regular walking intervention (T0) and post-intervention at 8 weeks (T1) to investigate the effects of the regular walking interventions.

#### 2.4.3. Measurements of Electrolyzed Water and Walking Group

Participants in the combined alkaline electrolyzed water and regular walking group were instructed to consume 2 L/day of supplemented alkaline electrolyzed water at pH 9.5 and to walk for at least 30 min five times/week or 150 min/week for 8 weeks. Research assistants controlled the frequency and duration of regular walking and alkaline electrolyzed water consumption reports 3 days/week and encouraged participants to adhere to the intervention protocol for safety management. Data were collected at the baseline before combined alkaline electrolyzed water and regular walking intervention (T0) and post-intervention at 8 weeks (T1) to investigate the effects of the combined alkaline electrolyzed water and regular walking interventions.

#### 2.4.4. Control Group

Control group participants were advised to continue their habitual diet and activity. They were recommended to not do any additional physical activity or to consume alkaline electrolyzed water supplementation, and research assistants controlled the control group participants’ daily activity and diet consumption 3 days/week and encouraged them to continue their habitual diet and activities for safety management. Data were collected at the baseline (T0) and post-intervention at 8 weeks (T1) to investigate the effects of habitual diet and activity.

### 2.5. Instruments and Measurements

Research assistants collected data using a questionnaire containing questions concerning participants’ demographic characteristics collected at the baseline, and included participant’s age, diabetes duration, gender, marital status, income, educational level and smoking status. Furthermore, factors including dietary habits, depression, anxiety and stress, that may influence oxidative stress, inflammatory markers and QoL, were also collected. This study measured the relevant biochemical features of T2DM, including AGEs, AOPPs, MDA, NLR, WBC and FBG.

#### 2.5.1. Measurements of Oxidative Stress and Inflammatory Markers

Participants were invited to participate in each clinical measurement session after 12 h of fasting. All venipuncture was performed by trained phlebotomists. All biochemical analysts and trained phlebotomists were blinded to the treatment group. Concentrations of AGEs [33], AOPPs [6] and MDA [34] were determined using enzyme-linked immunosorbent assays (ELISA; Bioassay Technology Laboratory, Shanghai, China) according to the manufacturer′s instructions, and the optical density was analyzed using an automated iMark^TM^ Microplate Absorbance Reader 1,681,130 (Bio-Rad, Hercules, CA, USA) at 450 nm at the Institute of Tropical Diseases, an International Research Center at the University of Airlangga (ISO 17025:2008 certified). According to the manufacturer’s instructions, the detection ranges of AGEs, AOPPs and MDA were 10~4000 ng/L, 0.1~40 ng/ml and 0.2~70 nmol/ml, respectively. The inter-assay and intra-assay coefficients of variation were <8% and <10%, respectively. The NLR and WBCs were measured using a fasting blood sample (3 mL) that was withdrawn from the antecubital vein. NLR was determined by dividing the absolute blood neutrophil count by the absolute lymphocyte count. The blood was mixed with dipotassium ethylenediaminetetraacetic acid (1.5~2.2 mg/mL). FBG, NLR and WBCs were analyzed using an automated XP-100 hematology cell counter (Sysmex, Kobe, Japan) in a private laboratory (ISO 14001: 2015 and 9001:2015 certified).

#### 2.5.2. Measurements of Quality of Life

The 36-Item Short Form Survey (SF-36) is one of the most widely used instruments to measure self-reported QoL [35]. In addition, the SF-36 is widely used in diabetes research [5,35]. The Indonesian-translated version of the SF-36 questionnaire had a test–retest reliability of *r* = 0.626–1 and had good internal consistency, with a Cronbach’s alpha of 0.789 [36]. The SF-36 encompasses eight subscales: role limitation (emotional), vitality, mental health, social functioning, general health, physical functioning, role limitation (physical) and bodily pain. These eight scales can be classified into three main domains, namely total QoL, a mental component score (MCS) and a physical component score (PCS). The MCS is the sum of the scores of role limitation (emotional), vitality, mental health and social functioning. The PCS is the sum of the scores of general health, physical functioning, role limitation (physical) and bodily pain. The total QoL is the sum of the mental and physical component scores. The total QoL, mental component score and physical component score range 0~100, with higher scores indicating a higher QoL [35].

#### 2.5.3. Measurements by a Food Frequency Questionnaire

To determine the behavioral dietary intake, 25 items of a self-reported food frequency questionnaire (FFQ) were used. Subjects were asked to report their average frequency intake of each food group, which has four options of more than once per day, one to six times per week, one to three times per month, and never. Moreover, we used the rank-determined dietary score (e.g., score 4: > once/day; score 3: 1–6 times/week; score 2: 1–3 times/month; score 1: never). The FFQ was established to have good performance validity with a content validity index of 0.75 [37], and was previously used in T2DM participants in Indonesia [38]. Moreover, we divided the cutoff values into two levels for each main content: carbohydrates (<17 and ≥17), protein (<18 and ≥18), fat (<20 and ≥20), fast foods (<9 and ≥9) and fiber (≥2 and <2).

#### 2.5.4. Measurements of Depression, Anxiety and Stress

We measured levels of depression, anxiety and stress using Depression, Anxiety and Stress Scale-21 (DASS-21) items [39]. Cronbach’s α values for the Indonesian version were 0.87, 0.85 and 0.72 for the depression, anxiety and stress subscales, respectively [40]. We divided DASS-21 scores into categorical data for anxiety (yes = score ≥ 8, no = score < 8), stress (yes = score ≥ 15, no = score <15), and depression (yes = score ≥ 10, no = score < 10).

#### 2.5.5. Measurements of Incidence of Adverse Events

The safety of the method of four parallel intervention groups was assessed by the incidence of adverse events during the trial study period. The potential adverse events included fatigue and increased urine production. These were self-recorded by participants and monitored by physicians once a week, as well as being well-documented by clinical nurses. In the event of a side-effect, the physician and investigators should immediately discontinue the intervention and intervene.

### 2.6. Study Fidelity

Study fidelity was established through meetings between investigators, clinical nurses, research assistants, research fellows and physicians to review the procedures, confirm competency measurement and equalize perceptions in the conducting of the research.

### 2.7. Statistical Analysis

Statistical analyses were performed using SPSS 25.0 (Chicago, IL, USA), with a *p* value of <0.05 considered statistically significant. Continuous and categorical data were reported using descriptive statistics of the mean (standard deviation (SD)) and *n* (%), respectively. Chi-squared and one-way analysis of variance (ANOVA) tests were respectively used to compare sociodemographic and baseline outcomes among the four groups. Generalized estimating equation models (GEE) with appropriate link functions and distribution assumptions were used to compare differential changes in outcomes across time and between groups. The models were adjusted for potential covariates including stress, anxiety and depression levels, and carbohydrate, protein, fat, fast food and fiber consumption scores. Missing data due to loss to follow-up (*n* = 1) were assumed to be missing at random, and data were analyzed following the intention-to-treat basis.

## 3. Results

Overall, 81 individuals with T2DM were randomized to AEW intervention (*n* = 20), regular walking intervention (*n* = 20), combined AEW and regular walking intervention (*n* = 20), and control group intervention (*n* = 21). The overall compliance of patients to the AEW and regular walking intervention was optimal. However, one control arm participant (1.24%) was lost to follow-up (contact lost) at 8 weeks, and no harms or unintended effects were measured in each group available for the intention-to-treat analysis. No statistically significant differences were noted in sociodemographic or clinical characteristics, including age, diabetes duration, gender, marital status, income, education, smoking status, physical activity, stress level, anxiety level, depression level or consumption of carbohydrates, protein, fat, fast food and fiber among the four groups (Table 1).

Moreover, Table 2 demonstrates that there were no significant differences in the baseline outcomes of AGEs, AOPP, MDA, blood glucose, the NLR, WBCs or QoL, including the mental component, physical component, and total QoL scores among the four groups (Table 2).

The univariate analysis of the primary outcome evaluation is presented in Table 3. Participants in the control group had increased levels of AGEs, AOPPs, MDA and WBCs, as well as a higher NLR, after 8 weeks of the study, but no significant changes in blood glucose. In the AEW, regular walking and combined AEW and walking groups there were significant reductions in AGEs, AOPPs, MDA, blood glucose and WBCs, as well as a lower NRL after the 8-week intervention (Table 3). 

In the walking group, after 8 weeks of the intervention, significant increases in QoL scores from before to after the intervention, with 95% confidence intervals (CIs), were detected, as follows: physical component score = 15.656 (12.966~18.346); mental component score = 13.869 (10.397~17.341); total QoL = 14.763 (12.364~17.163). The physical component and total QoL scores significantly increased in the AEW group (11.938 (9.687~14.118) and 6.315 (5.108~7.523) respectively), but no changes occurred in the mental component score (0.694 (−0.090~1.447)). Moreover after 8 weeks of the intervention, the combined AEW water and regular walking group saw significant increases in the physical component, mental component and total QoL scores of 22.750 (20.242~25.258), 27.113 (24.089~30.163), and 24.933 (23.090~26.776), respectively. In contrast, the control group showed no changes in the physical component, mental component or total QoL scores (Table 4).

The intervention effects on biomarkers and QoL after the 8-week intervention are shown in Table 5 and Table 6. There were significant within-time-induced differences in AGEs, MDA, the NLR and WBCs before and after the 8-week intervention, but no significant within-time-induced differences in AOPPs, blood glucose or QoL parameters before or after the 8 weeks. Moreover, there were no differences in any biomarkers and QoL parameters between each intervention group and the control group at the baseline. After full adjustment, the significance of the interaction group x time analysis for all biomarkers revealed that participants in the AEW group exhibited significant reductions in AOPPs (ß = −0.185; 95% CI = −0.302~−0.068), AGEs (ß = −0.298; 95% CI = −0.433~−0.163), MDA (ß = −0.566; 95% CI = −0.691~−0.440), blood glucose (ß = −54.583; 95% CI = −96.085~−13.080), the NLR (ß = −1.666; 95% CI = −1.771~−1.560) and WBCs (ß = −2.406; 95% CI = −2.687~−2.125), but increased in SF36-PCS (ß = 14.034; 95% CI = 10.727~17.342) and SF36-total QoL scores (ß = 6.828; 95% CI = 4.099~9.557) after the 8-week intervention compared to the control group. Besides, participants in the regular walking group had significant declines in AOPPs (ß = −0.184; 95% CI = −0.313~−0.055), AGEs (ß = −0.304; 95% CI = −0.445~−0.163), MDA (ß = −0.579; 95% CI = −0.727~−0.431), blood glucose (ß = −49.680; 95% CI = −92.196~−13.080), the NLR (ß = −1.623; 95% CI = −1.713~−1.533) and WBCs (ß = −2.020; 95% CI = −2.289~−1.752), but increased in SF36-PCS (ß = 18.017; 95% CI = 14.170~21.865), SF36-MCS (ß = 13.360; 95% CI = 7.522~19.198) and SF36-total QoL scores (ß = 15.689; 95% CI = 11.964~19.414) after the 8-week intervention compared to the control group. Moreover, compared to the control group, participants in the combined AEW with regular walking group also had significant reductions in AOPPs (ß = −0.264; 95% CI = −0.454~0.074), AGEs (ß = −0.364; 95% CI = −0.546~−0.182), MDA (ß = −0.716; 95% CI = −0.909~−0.524), blood glucose (ß = −57.223; 95% CI = −99.111~−7.163), the NLR (ß = −1.798; 95% CI = −1.897~−1.897) and WBCs (ß = −2.833; 95% CI = −3.119~−2.547), but higher SF36-PCS (ß = 24.483; 95% CI = 20.881~28.805), SF36-MCS (ß = 25.649; 95% CI = 20.310~30.988) and SF36-total QoL scores (ß = 25.068; 95% CI = 21.961~28.175) after the 8-week intervention (Table 5 and Table 6).

## 4. Discussion

To our knowledge, this is the first community-based randomized control trial study to explore the effects of regular walking and consumption of AEW on oxidative stress, inflammatory markers and QoL among patients with T2DM. Community-based health programs are practical and relatively low-resource, often requiring lifestyle-changing educational programs. Previous studies suggested that community-based RCT can diagnose as well as monitor how well diabetes is controlled and improve community health habits, which ultimately reinforces the high priority given to the translation of research into new practices [41].

These results extend our understanding that T2DM participants who regularly walked for a duration of 30 min at least 5 times/week, or spent 150 min/week regularly walking for 8 weeks, exhibited significantly reduced WBCs, NLR, FBG, MDA, AGEs and AOPPs. Our study was consistent with a prospective cohort study, which indicated that low/moderate-intensity physical activity such as walking was inversely associated with WBC counts, especially for patients with moderate to vigorous physical activity [42]. Additionally, physical activity of three or more times/week decreased the NLR by 0.07-fold compared to physical inactivity [14]. Engaging in 30 min of moderate-intensity walking contributed to strengthening their antioxidant defenses by increasing their antioxidant enzymes by 13% [43]. Additionally, an animal study [13] investigated the effect of low physical activity with regular walking on a treadmill once a day for four consecutive weeks on oxidative stress in cardiac tissues in rats with streptozotocin-induced diabetes, and their main finding was that regular walking on a treadmill decreased cardiac oxidative stress, as shown by lower levels of MDA. We also found that regular walking by diabetic patients significantly decreased MDA levels. Moreover, our results are consistent with prior research [44,45], which recommends that regular and moderate-intensity physical exercise is able to control glycemia and maintain blood pressure, thus reducing complication risks, by preventing pathophysiological mechanisms at different levels, including lowering oxidative stress and inflammation levels, which are major features of diabetes. However, very few studies have focused on investigating the effects of regular walking on the formation of AGEs [12] and AOPPs [11] in patients with T2DM. It can be assumed that regular walking will reduce the levels of AGEs and AOPPs, which as we know are caused by hyperglycemia [46]. Therefore, improvements in glycemic control in T2DM patients with physical activity are linked to a reduction in peripheral resistance to insulin [47]. This condition could lead to the attenuation of the formation and accumulation of AGEs and AOPPs. Moreover, we revealed that a regular walking intervention is a possible treatment to help T2DM patients in terms of improving their QoL. Our findings were in agreement with a previous study which showed that physical activity was the strongest predictor of the mental component score and physical component score, after adjusting for the body-mass index, age and sex among patients with T2DM [48]. Our study suggested that regular walking for at least 30 min each time and five times/week may promote a healthy lifestyle for controlling inflammatory markers and oxidative stress, as well as improving the QoL, which is particularly needed among Indonesians who are usually physically inactive.

In the present study, we found that the AEW group had lower oxidative stress and inflammatory markers after an 8-week intervention compared to the control group. The antioxidant, anti-obesity and antidiabetic effects of AEW were reported in previous studies [49]. The protective mechanism of AEW is attributed to its active atomic hydrogen that has a high reducing ability, can contribute to ROS-scavenging activity, and may participate in redox reactions, thus increasing levels of antioxidants [21]. Active atomic hydrogen is produced at the cathode during the electrolysis of water, resulting in molecular hydrogen and hydroxide ions. This condition illustrates the high power of hydrogen to make water more alkaline, and AEW has extremely high dissolved molecular hydrogen properties [18]. Hydrogen can rapidly diffuse across cell membranes, and this may provide powerful protection against oxidative stress through its ability to bind with hydroxyl radicals [50]. In addition, inflammation is a process known to be closely linked with oxidative stress, and several studies addressed the potential anti-inflammatory effects of hydrogen. Hydrogen treatment can reduce the interleukin-6 and tumor necrosis factor, as well as other proinflammatory molecules such as interleukin-12 and interferon gamma [51]. In our study, we used easy-to-detect and inexpensive inflammatory markers in humans, and our results showed that AEW could reduce WBCs and the NLR after 8 weeks of treatment compared to the control group. These results confirm the potential anti-inflammatory effect of AEW in humans. In terms of AOPPs, AGEs and MDA, our study found that after 8 weeks of treatment, AEW had significantly reduced levels of AOPPs, AGEs and MDA. This may be attributable to the potential effect of AEW of decreasing blood glucose in T2DM [52]. Additionally, our study also showed that the physical component score and total QoL of people consuming AEW for an 8-week period increased compared to the control group. Previous studies reported that ingesting AEW containing hydrogen, which is considered an antioxidant substance, may reduce oxidative stress, thus allowing for improved sleep quality. Therefore, good-quality sleep can lead to better physical activity by helping to reduce fatigue, ensure the recovery of endurance, and improve overall physical performance [53], which may produce better total QoL and physical component scores.

Additionally, our study demonstrated that participants in the combined regular walking and AEW intervention groups exhibited synergistic amelioration of FBG, AOPPs, AGEs, MDA, NLR and WBCs compared to a control group after an 8-week intervention. The possible mechanism for this synergistic effect might be the effects on antioxidants. Those findings are similar to the results of this study, which determined that antioxidative stress rose in those who consumed AEW. Importantly, Weidman et al. [54] stated that the consumption of AEW may exert positive effects by increasing antioxidant defense and reducing the basal production of oxidants after exercise training. Moreover, it was found that AEW has beneficial effects on exercise due to its free radical-scavenging properties, thereby enabling the maintenance of muscle performance and redox homeostasis during consecutive days of exercise [29]. Furthermore, a previous randomized and double-blind study reported that AEW significantly restored rehydration and reduced high-shear blood viscosity by an average of 6.30% during a 2 h recovery period, compared to standard mineral water. Thus, blood viscosity disruption promotes higher oxidative stress in which ROS can accumulate, which are capable of producing their own free radicals [54]. Our findings also support our secondary hypothesis that the combined AEW and regular walking group was significantly associated with a higher QoL as assessed by SF-36 scores, including PCS, MCS and total QoL scores, which indicates a good performance in the areas of physical and mental condition. This highlights that antioxidants together with physical activity may have positive effects on reducing clinical damage due to decreased blood glucose, inflammation and oxidative stress, and diminishing the development of complications induced by oxidative stress. These conditions lead to better glycemic control in patients with T2DM [3]. On the other hand, management of hyperglycemia can delay or prevent complications and optimize the QoL [31]. This mechanism might provide convincing insights into pathways that influence both mental and physical QoL in patients with T2DM.

Our study had some limitations that must be considered. First, although it is essential to determine the immediate effects after an 8-week intervention, evaluation of the long-term follow-up is also necessary. Second, walking speed was not recorded, and this could have induced an underestimation of the effect of the regular walking intervention on outcomes. Third, we did not measure the pharmacokinetics of AEW in the gastrointestinal region of the subjects. Therefore, future research should investigate the molecular mechanisms of hydrogen at the physiological level, such as the gastrointestinal region. Fourth, the present study was not able to measure HbA1c due to the higher cost and longer time required to detect (>3 months), and did not include dietary management intervention, which may influence our findings. Lastly, all enrolled participants were of the Javanese ethnicity; consequently, the generalizability of our result may be limited.

## 5. Conclusions

In conclusion, our findings revealed that regular walking and the drinking of AEW, both independently and synergistically, reduced FBG, AGEs, AOPPs, MDA, WBCs and the NLR, as well as promoting the QoL of patients with T2DM over 8 weeks, so it is a feasible treatment in advanced T2DM.

## Figures and Tables

**Figure 1 antioxidants-09-00946-f001:**
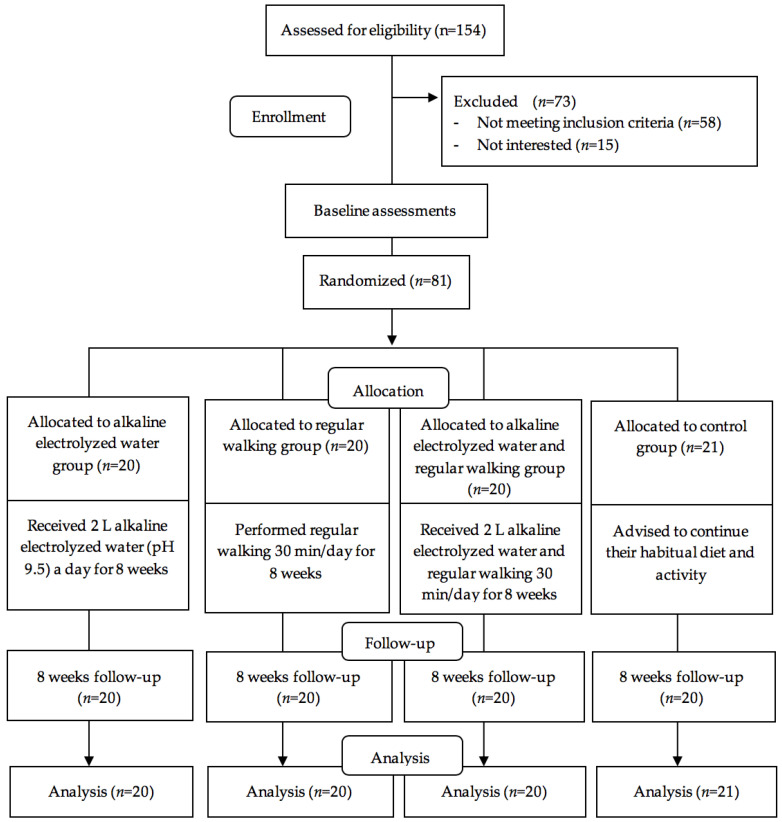
CONSORT diagram of this Study.

**Table 1 antioxidants-09-00946-t001:** Comparisons of participants’ sociodemographic and clinical data among the four groups (*n* = 81).

Sociodemographic and Clinical Data	AEW Group (*n* = 20), *n* (%)	Walking Group (*n* = 20), *n* (%)	AEW and Walking Group (*n* = 20), *n* (%)	Control Group (*n* = 21), *n* (%)	*p* Value (*x*^2^) ^a^
Age (years), (mean, SD)	57.50 (5.48)	54.70 (4.87)	56.15 (4.96)	55.71 (4.97)	0.359 (1.089) ^b^
Diabetes duration (years), (mean, SD)	4.30 (1.87)	4.25 (1.62)	4.20 (1.51)	4.19 (1.54)	0.996 (0.019) ^b^
Gender					
Female	11 (55.0)	13 (65.0)	13 (65.0)	13 (61.9)	0.904 (0.565)
Male	9 (45.0)	7 (35.0)	7 (35.0)	8 (38.1)	
Marital status					
Married	11 (55.0)	9 (45.0)	11 (55.0)	13 (61.9)	0.754 (1.194)
Non-married	9 (45.0)	11 (55.0)	9 (45.0)	8 (38.1)	
Income (IDR)					
Low income	8 (40.0)	10 (50.0)	10 (50.0)	12 (57.1)	0.749 (1.216)
High income	12 (60.0)	10 (50.0)	10 (50.0)	9 (42.9)	
Education					
ISCED < 3	9 (45.0)	8 (40.0)	9 (45.0)	12 (57.1)	0.723 (1.325)
ISCED ≥ 3	11 (55.0)	12 (60.0)	11 (55.0)	9 (42.9)	
Smoking status					
Smoker	7 (35.0)	5 (25.0)	6 (30.0)	7 (33.3)	0.961 (1.484)
Secondhand smoking	7 (35.0)	9 (45.0)	10 (50.0)	9 (42.9)	
Non-smoker	6 (30.0)	6 (30.0)	4 (20.0)	5 (23.8)	
DASS-21 stress					
Yes (≥15 score)	12 (60.0)	12 (60.0)	11 (55.0)	12 (57.1)	0.986 (0.146)
No (<15 score)	8 (40.0)	8 (40.0)	9 (45.0)	9 (42.9)	
DASS-21 anxiety					
Yes (≥8 score)	11 (55.0)	11 (55.0)	11 (55.0)	12 (57.1)	0.999 (0.029)
No (<8 score)	9 (45.0)	9 (45.0)	9 (45.0)	9 (42.9)	
DASS-21 depression					
Yes (≥10 score)	12 (60.0)	10 (50.0)	11 (55.0)	12 (57.1)	0.933 (0.434)
No (<10 score)	8 (40.0)	10 (50.0)	9 (45.0)	9 (42.9)	
Carbohydrate consumption score					
<17	10 (50.0)	9 (45.0)	9 (45.0)	12 (57.1)	0.846 (0.816)
≥17	10 (50.0)	11 (55.0)	11 (55.0)	9 (42.9)	
Protein consumption score					
<18	10 (50.0)	8 (40.0)	8 (40.0)	11 (52.4)	0.789 (1.051)
≥18	10 (50.0)	12 (60.0)	12 (60.0)	10 (47.6)	
Fat consumption score					
<20	10 (50.0)	9 (45.0)	9 (45.0)	10 (47.6)	0.987 (0.140)
≥20	10 (50.0)	11 (55.0)	11 (55.0)	11 (52.4)	
Fast food consumption score					
<9	12 (60.0)	8 (40.0)	7 (35.0)	12 (57.1)	0.293 (3.723)
≥ 9	8 (40.0)	12 (60.0)	13 (65.0)	9 (42.9)	
Fiber consumption score					
≥2	9 (45.0)	7 (35.0)	8 (40.0)	12 (57.1)	0.521 (2.256)
<2	11 (55.0)	13 (65.0)	12 (60.0)	9 (42.9)	

Note—*n*: number, AEW: alkaline electrolyzed water, SD: standard deviation, IDR: Indonesian rupiah rate, ISCED: international standard classification of education, DASS-21: Depression, Anxiety, and Stress Scale—21 items. ^a^ Chi-square statistics, with the *x*^2^-value in the bracket behind *p*-value. ^b^ Analysis of variance, with the *F*-value in the bracket behind *p*-value.

**Table 2 antioxidants-09-00946-t002:** Baseline values of participants’ oxidative stress, inflammatory biomarkers and quality of life among the four groups (*n* = 81).

Sociodemographic and Clinical Data	AEW Group (*n* = 20), Mean (SD)	Walking Group (*n* = 20), Mean (SD)	AEW and Walking Group (*n* = 20), Mean (SD)	Control Group (*n* = 21), Mean (SD)	*p* Value (*F*) ^a^
AOPPs (ng/ml)	0.38 (0.57)	0.35 (0.43)	0.53 (0.70)	0.44 (0.65)	0.794 (0.343)
AGEs (ng/L)	0.35 (0.54)	0.27 (0.32)	0.46 (0.59)	0.39 (0.58)	0.715 (0.454)
MDA (nmol/ml)	0.77 (0.42)	0.85 (0.69)	0.97 (0.67)	0.96 (0.62)	0.691 (0.489)
FBG (mg/dl)	298.95 (48.84)	303.95 (67.95)	324.60 (59.63)	287.24 (80.43)	0.331 (1.159)
WBCs (10^3^/µL)	7.92 (0.51)	8.19 (0.44)	8.48 (0.93)	8.19 (1.16)	0.199 (1.589)
NLR	2.63 (0.22)	2.55 (0.18)	2.61 (0.22)	2.62 (0.23)	0.691 (0.488)
SF36-PCS	45.16 (6.34)	45.16 (6.34)	44.94 (6.82)	43.99 (5.75)	0.922 (0.161)
SF36-MCS	46.52 (8.12)	47.50 (9.01)	46.43 (6.67)	46.66 (8.99)	0.976 (0.070)
SF36-Total QoL	45.84 (6.22)	46.33 (6.63)	45.69 (4.64)	45.33 (5.20)	0.955 (0.107)

Note—*n*: number, SD: standard deviation, AEW: alkaline electrolyzed water, AOPPs: advanced oxidation protein products, AGEs: advanced glycation end products, MDA: malondialdehyde, FBG: fasting blood glucose, WBCs: white blood cells, NLR: neutrophil-lymphocyte ratio, SF-36: Short Form 36, PCS: Physical component score, MCS: mental component score, QoL: quality of life. ^a^ Analysis of variance, with the F-value in the bracket behind *p*-value of <0.01.

**Table 3 antioxidants-09-00946-t003:** Oxidative stress and inflammatory biomarkers before and after the intervention in participants with type 2 diabetes mellitus (*n* = 81).

Variables	AEW Group (n = 20), Mean (SD)			Walking Group (n = 20), mean (SD)			AEW and Walking Group (n = 20), Mean (SD)			Control Group (n = 21), Mean (SD)		
	Baseline	8-Week	Diff (95% CI), *p* Value	Baseline	8-Week	Diff (95% CI), *p* Value	Baseline	8-Week	Diff (95% CI), *p* Value	Baseline	8-Week	Diff (95% CI), *p* Value
AOPPs (ng/ml)	0.38 (0.57)	0.28 (0.47)	−0.101 (−0.151~−0.045) **	0.35 (0.43)	0.26 (0.32)	−0.089 (−0.145~−0.322) **	0.53 (0.70)	0.30 (0.42)	−0.225 (−0.369~−0.080) **	0.44 (0.65)	0.54 (0.67)	0.83 (0.001~0.165) *
AGEs (ng/L)	0.35 (0.54)	0.27 (0.47)	−0.077 (−0.121~−0.029) **	0.27 (0.32)	0.20 (0.26)	−0.072 (−0.110~−0.034) **	0.46 (0.59)	0.28 (0.37)	−0.180 (−0.287~−0.073) **	0.39 (0.58)	0.60 (0.80)	0.218 (0.101~0.335) **
MDA (nmol/ml)	0.77 (0.42)	0.46 (0.41)	−0.312 (−0.352~−0.272) **	0.85 (0.69)	0.53 (0.58)	−0.321 (−0.386~−0.256) **	0.97 (0.67)	0.47 (0.45)	−0.499 (−0.628~−0.370) **	0.96 (0.62)	1.21 (0.75)	0.253 (0.154~0.351) **
FBG (mg/dl)	298.95 (48.84)	278.00 (49.03)	−20.950 (−24.470~−17.430) **	303.95 (67.95)	288.00 (67.70)	−15.950 (−16.637~−15.263) **	324.60 (59.63)	297.85 (58.89)	−26.750 (−27.697~−25.803) **	287.24 (80.43)	331.19 (124.16)	43.952 (−5.688~93.592)
WBCs (103/µL)	7.92 (0.51)	5.70 (0.51)	−2.220 (−2.473~−1.966) **	8.19 (0.44)	6.34 (0.55)	−1.855 (−2.064~−1.645) **	8.48 (0.93)	5.86 (0.78)	−2.620 (−2.832~−2.407) **	8.19 (1.16)	8.42 (1.12)	0.233 (0.121~0.346) **
NLR	2.63 (0.22)	1.39 (0.29)	−1.235 (−1.297~−1.173) **	2.55 (0.18)	1.36 (0.18)	−1.193 (−1.200~−1.186) **	2.61 (0.22)	1.21 (0.16)	−1.392 (−1.437~−1.347) **	2.62 (0.23)	3.09 (0.30)	0.471 (0.329~0.612) **

Note—*n*: number, SD: standard deviation, Diff: difference between after and before, CI: confidence interval, AEW: alkaline electrolyzed water, AOPPs: advanced oxidation protein products, AGEs: advanced glycation end products, MDA: malondialdehyde, FBG: fasting blood glucose, WBCs: white blood cells, NLR: neutrophil-lymphocyte ratio. One-way analysis of variance (ANOVA) with * *p* < 0.05; ** *p* < 0.001.

**Table 4 antioxidants-09-00946-t004:** Quality of life before and after the intervention in participants with T2DM (*n* = 81).

Variables	AEW Group (*n* = 20), Mean (SD)			Walking Group (*n* = 20), Mean (SD)			AEW and Walking Group (*n* = 20), Mean (SD)			Control Group (*n* = 21), mean (SD)		
	Baseline	8-Week	Diff (95% CI), *p* Value	Baseline	8-Week	Diff (95% CI), *p* Value	Baseline	8-Week	Diff (95% CI), *p* Value	Baseline	8-Week	Diff (95% CI), *p* Value
SF36- PCS	45.16 (6.34)	57.09 (4.84)	11.938 (9.687~14.188) **	45.16 (6.34)	60.81 (3.24)	15.656 (12.966~18.346) **	44.94 (6.82)	67.69 (3.10)	22.750 (20.242~25.258) **	43.99 (5.75)	42.11 (6.57)	−1.875 (−4.107~0.357)
SF36- MCS	46.52 (8.12)	47.21 (7.89)	0.694 (−0.090~1.447)	47.50 (9.01)	61.37 (12.53)	13.869 (10.397~17.341) **	46.43 (6.67)	73.54 (5.81)	27.113 (24.089~30.136) **	46.66 (8.99)	47.84 (12.51)	1.183 (−2.602~4.967)
SF36-Total QoL	45.84 (6.22)	52.16 (5.38)	6.315 (5.108~7.523) **	46.33 (6.63)	61.09 (7.15)	14.763 (12.364~17.163) **	45.69 (4.64)	70.62 (3.25)	24.933 (23.090~26.776) **	45.33 (5.20)	44.98 (8.33)	−0.346(−2.602~1.909)

Note—*n*: number participants, SD: standard deviation, Diff: difference between after and before, CI: confidence interval, AEW: alkaline electrolyzed water, SF-36: Short Form 36, PCS: physical component score, MCS: mental component score, QoL: quality of life. One-way analysis of variance (ANOVA) with ** *p* < 0.001.

**Table 5 antioxidants-09-00946-t005:** Evaluation of the intervention on T2DM-associated inflammatory biomarkers based on a general estimating equation analysis (*n* = 81).

Variable	Within-time Ref: Baseline ß (95% CI), *p* Value	Between Group			Interaction Group (AEW) × Time, Ref: (CG) × Time ß (95% CI), *p* Value	Interaction Group (Walking) × Time, Ref: (CG) × Time ß (95% CI), *p* Value	Interaction Group (both AEW and Walking) × Time, Ref: (CG) × Time ß (95% CI), *p* Value
		AEW vs. CG ß (95% CI), *p* Value	Walking vs. CG ß (95% CI), *p* Value	AWE and Walking vs. CG ß (95% CI), *p* Value			
AOPPs (ng/ml)	0.081 (−0.020~0.182)	−0.017 (−0.392~0.359)	−0.151 (−0.482~0.181)	0.020 (−0.355~0.395)	−0.185 (−0.302~−0.068) *	−0.184 (−0.313~−0.055) *	−0.264 (−0.454~0.074) *
AGEs (ng/L)	0.214 (0.094~0.333) **	0.006 (−0.338~0.350)	−0.157 (−0.444~0.130)	0.028 (−0.305~0.362)	−0.298 (−0.433~−0.163) **	−0.304 (−0.445~−0.163) **	−0.364 (−0.546~−0.182) **
MDA (nmol/ml)	0.253 (0.144~0.362) **	−0.138 (−0.467~0.191)	−0.152 (−0.524~0.220)	−0.040 (−0.400~0.321)	−0.566 (−0.691~−0.440) **	−0.579 (−0.727~−0.431) **	−0.716 (−0.909~−0.524) **
FBG (mg/dl)	36.878 (−2.639~76.396)	10.640 (−15.734~37.013)	3.200 (−21.894~28.294)	20.499 (−10.877~51.874)	−54.583 (−96.085~−13.080) *	−49.680 (−92.196~−13.080) *	−57.223 (−99.111~−7.163) *
WBCs (10^3^/µL)	0.234 (0.098~0.369) **	−0.395 (−0.977~0.187)	−0.032 (−0.601~0.538)	0.269 (−0.404~0.941)	−2.406 (−2.687~−2.125) **	−2.020 (−2.289~−1.752) **	−2.833 (−3.119~−2.547) **
NLR	0.440 (0.353~0.526) **	0.025 (−0.101~0.152)	−0.091 (−0.221~0.039)	−0.034 (−0.163~0.095)	−1.666 (−1.771~−1.560) **	−1.623 (−1.713~−1.533) **	−1.798 (−1.897~−1.897) **

Note—*n*: number of participants, ß: regression coefficient, CI: confidence interval, AEW: alkaline electrolyzed water, CG: control group, AOPPs: advanced oxidation protein products, AGEs: advanced glycation end products, MDA: malondialdehyde, FBG: fasting blood glucose, WBCs: white blood cells, NLR: neutrophil-lymphocyte ratio. ß values and 95% CIs were estimated using generalized estimating equations after adjusting for stress, anxiety and depression levels, and carbohydrate, protein, fat, fast food and fiber consumption scores with * *p* < 0.05; ** *p* < 0.001.

**Table 6 antioxidants-09-00946-t006:** Evaluation of the intervention on the quality of life in participants with T2DM based on general estimating equation analysis (*n* = 81).

Variable	Within-Time Ref: Baseline ß (95% CI), *p* Value	Between Group			Interaction Group (AEW) × Time, Ref: (CG) × Time ß (95% CI), *p* Value	Interaction Group (Walking) × Time, Ref: (CG) × Time ß (95% CI), *p* Value	Interaction Group (both AEW & walking) × Time, Ref: (CG) × Time ß (95% CI), *p* Value
		AEW vs. CGß (95% CI), *p* Value	Walking vs. CGß (95% CI), *p* Value	Both vs. CGß (95% CI), *p* Value			
SF36- PCS	−1.794 (−4.301~0.714)	1.366 (−2.335~5.068)	0.908 (−2.892~4.708)	0.670 (−3.174~4.514)	14.034 (10.727~17.342) **	18.017 (14.170~21.865) **	24.483 (20.881~28.085) **
SF36- MCS	1.753 (−2.526~6.032)	0.505 (−4.082~5.092)	1.604 (−3.428~6.637)	0.782 (−3.969~5.533)	−0.378 (−4.846~4.089)	13.360 (7.522~19.198) **	25.649 (20.310~30.988) **
SF36-Total QoL	−0.021 (−2.463~2.422)	0.936 (−2.223~2.422)	1.257 (−2.225~4.738)	0.727 (−2.332~3.786)	6.828 (4.099~9.557) **	15.689 (11.964~19.414) **	25.068 (21.961~28.175) **

Note: *n*: number of participants, ß: regression coefficient, CI: confidence interval, AEW: alkaline electrolyzed water, CG: control group, SF−36: Short Form 36, PCS: Physical component score, MCS: Mental component score, QoL: quality of life. ß values and 95% CIs were estimated using generalized estimating equations after adjusting for stress, anxiety, and depression levels, and carbohydrate, protein, fat, fast food and fiber consumption scores with ** *p* < 0.001.

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
