# Peer review of "Synergistic Effects of Regular Walking and Alkaline Electrolyzed Water on Decreasing Inflammation and Oxidative Stress, and Increasing Quality of Life in Individuals with Type 2 Diabetes: A Community Based Randomized Controlled Trial"

_antioxidants, 2020, doi:10.3390/antiox9100946_

Round 1

Reviewer 1 Report

The study is nicely and pricisely described. Especially the Methods en results section are desribed in full detail. With respect to the Introduction and Discussion more information is needed to what and why this study is of interest/importance. 

Please make clear in the introduction why you want te combine these two interventions, walking and drinking AEW. What is the rational  for this, it seems random and is not explained yet.

Also state why these biomarkers will be discussed in relation to Quality of Life. That is not mentioned clearly.

How come the study was conducted in Indonesia by Taiwanese people? Add the reason for this in the method section. Is it due to effeciency (costs) or due to disease relevance? 

No information is available about medication of the subjects. Were they all on T2D medication? How was this distributed in the four groups? Did subjects remain on medication during treatment? Were different medicines allowed? Please add information because this will affect blood parameters too. 

line 160: the water was produced fresh twice a week; what did this mean for participants, had they collect the water twice a week too? Please add. 

line 203: What is 12 hours of rapid? Do you mean 12 hours of fasting?

What is known about compliance of the subjects in the different intervention groups? No information is provided in the results while this is key. When all are not compliant you can not expect anything. Please mention how well subjects adhered to their treatment interventions. 

ln 211-223: please refer to the method reference in stead of this detailed method description.

ln 240-241: role emotional and role physical is in other publications refered to as role limitation - emotional or role limitation - physical.

Ln 259/260: Do you mean grams or E%, not clear; please add.

It seems like statistics has been done one by one; so each group versus control for the intervention. Why is this not done in one model (ANOVA) like was done for baseline? And perform post-hoc tests to examine where differences exist?

Of some Tables the lines are not straight and it becomes difficult to read and compare data correctly. 

line 318: The order is changed; please mention AEW first for clarity. It is nice when this is consistent along the whole paper.

It is not easy to read line 335-365 and Table 5 and 6. Remove duplicate information. Is description in results section possible with the data in Table 5 and 6?

Is it possible that the hydrogen has still special properties after consumption? Is this not lost in the stomach in the interaction with food and drinks in the stomach?

The Discussion is a bit messy with respect to information; it jumps from topic to topic, not easy to follow. Combine parts that discuss the same topic (e.g. antioxidants in line 411-415 and in line 444-447).

Ln 429-437 all kind of topics; try to focus for a better/ more effective discussion of the results.  

ln 458: increased blood glucose? Don't you mean decreased?

The results of the SF-36 are not functionally explained; felt subjects better? What did subjects report? Did this affect their compliance? The relation with the inflammation markers is not so logic, so please elaborate on that.

Were physical effects found? Did you measure? Can you refer to that paper or mention main results? 

In the conclusion you come up with the relevance of the interventions to be advised by health care professionals. However, in the paper no discussion is present about the community set-up (why is this so special) and are nurses able to do provide this information (time, interaction, education) to patients. 

Please read the text once more carefully because some typing errors and not completely correct sentences are present in the text. Some plural forms are mentioned when a single form is needed etc.

Author Response

RE: antioxidants-929041-Version 1

Response to Reviewer 1 Comments

Point 1. The study is nicely and precisely described. Especially the methods and results section are described in full detail. With respect to the Introduction and Discussion more information is needed to what and why this study is of interest/importance. 

Response 1: Thank you very much. We appreciate this reviewer’s comments. In this revised manuscripts, we added a description to make a clear the importance of this study as follows in the section of the introduction and discussion of our study.

“This highlight that regular walking together with AEW may have positive effects on reducing inflammation and oxidative stress, which consequently increase QoL in individual with T2DM. However, these relationships require clarification” (Please see line 103–106 on page 3).

“This highlights that antioxidants together with physical activity may have positive effects on reducing clinical damage due to increased blood glucose, inflammation, oxidative stress, and diminishing the development of complications induced by oxidative stress. These conditions lead to better glycemic control in patients with T2DM [3]. On the other hand, management of hyperglycemia can delay or prevent complications and optimize the QoL [31]. This mechanism might provide convincing insights into pathways that influence both mental and physical QoL in patients with T2DM” (Please see line 437–443 on page 16).

Point 2. Please make clear in the introduction why you want the combine these two interventions, walking and drinking AEW. What is the rational for this, it seems random and is not explained yet. Also state why these biomarkers will be discussed in relation to Quality of Life. That is not mentioned clearly.

Response 2: Thank you for your valuable suggestion. We appreciate this reviewer’s comments. In this revised manuscripts, we added above point of “Interestingly, it was found that AEW has beneficial effects on exercise due to its free radical-scavenging properties, thereby enabling the maintenance of muscle performance and redox homeostasis during consecutive days of exercise [29]. This highlight that regular walking together with AEW may have positive effects on reducing inflammation and oxidative stress, which consequently increase QoL in individual with T2DM. However, these relationships require clarification” (Please see line 101–106 on page 3).

“Also, a previous study showed that treatment with 2 L/day of AEW for patients receiving radiotherapy for liver tumors improved QoL within 6 weeks [28]. Moreover, lower levels of inflammation such as NLR and WBCs were positively associated with higher QoL [5]. Similarly, amelioration of chronic inflammation is a potentially novel biobehavioral therapeutic strategy for improving QoL [10]” (Please see line 95–100 on page 3).

Point 3. How come the study was conducted in Indonesia by Taiwanese people? Add the reason for this in the method section. Is it due to efficiency (costs) or due to disease relevance? 

Response 3: Thank you for your comments and valuable suggestion. Let us clarify and revise this point. The first authors ”Yohanes Andy Rias” is from Indonesia. He is an investigator and project administration as well as a PhD Candidate, at The School of Nursing, College of Nursing, Taipei Medical University, Taiwan.

Point 4. No information is available about medication of the subjects. Were they all on T2D medication? How was this distributed in the four groups? Did subjects remain on medication during treatment? Were different medicines allowed? Please add information because this will affect blood parameters too.

Response 4: Thank you for your valuable suggestion. In this revised manuscripts, we added a description to make a clear in the inclusion criteria that all patients were on stable oral hypoglycemic agents (Metformin and/or Glibenclamide). Moreover, we excluded patients who were on insulin injection. All the medication were distributed by the physician and no changes were allowed (Please see line 145–146; and 149–152 on page 4).

Point 5. line 160: The water was produced fresh twice a week; what did this mean for participants, had they collect the water twice a week too? Please add. 

Response 5: Thank you for your valuable comments and suggestion. In this revised manuscripts, we have added above the point based on reviewer’s suggestion as follows of “We would like to confirm that all patients had collected the AEW twice a week” (Please see line 167–168 on page 5).

Point 6. line 203: What is 12 hours of rapid? Do you mean 12 hours of fasting?

Response 6: Thank you for your valuable suggestion. We have revised and corrected the word of “rapid” to “fasting” (Please see line 211 on page 6).

Point 7. What is known about compliance of the subjects in the different intervention groups? No information is provided in the results while this is key. When all are not compliant you can’t expect anything. Please mention how well subjects adhered to their treatment interventions. 

Response 7: Thank you for your valuable suggestion. In this revised manuscripts, we added a description to make a clear the compliance of the participants in the different interventions based on the reviewer’s suggestion as follows in the section of the result of our study (Please see line 279–281 on page 7).

“Overall compliance patients to the AEW and regular walking intervention was optimal. However, one control arm participants (1.24%) were lost in contact at 8 weeks and no harms and unintended effect in each group, available for the intention-to-treat analysis”.

Point 8. ln 211-223: please refer to the method reference instead of this detailed method description.

Response 8: Thank you for your valuable comments. In order to make data to be better presented, we reorganized and refer to the method references based on reviewer’s suggestion (Please see line 211–220 on page 6).

Point 9. ln 240-241: role emotional and role physical is in other publications referred to as role limitation - emotional or role limitation - physical.

Response 9: Thank you for your valuable comments. We re-word “role emotional and role physical” to “role limitation - emotional and role limitation - physical” follow reviewer’s suggestion (Please see line 231, 232, 234, and 236 on page 6).

Point 10. Ln 259/260: Do you mean grams or E%, not clear; please add.

Response 10: Thank you for your comments. We included more details about the specification of the food frequency questionnaire. Subjects were asked to report their average frequency intake of each food group, which has four options of more than once per day, one to six times per week, one to three times per month, and never. Moreover we used rank determine dietary score ( e.g. score 4: > once/day; score 3: 1-6 times/week; score 2: 1-3 times/month; score 1: never) (Please see line 241–244 on page 6–7).

“……Moreover, we divided the cutoff values into two levels for each main content: carbohydrates (<17 and ≥17), protein (<18 and ≥18), fat (<20 and ≥20), fast foods (<9 and ≥9), and fiber (≥2 and <2), which is no value were added. Those refer to reviewer’s suggestion in this revised manuscript (Please see line 246–248 on page 7).

Point 11. It seems like statistics has been done one by one; so each group versus control for the intervention. Why is this not done in one model (ANOVA) like was done for baseline? And perform post-hoc tests to examine where differences exist?

Response 11: Thank you for your comments. We would like to confirm that our statistical analysis was used by one-way ANOVA (table 3 and table 4 on page 10 and 12). Only baseline characteristics which is categorical variables used chi-square test (table 1 on page 8). Moreover, GEE to examine the differences between time and groups to examine where differences exist (table 5 and 6 on page 13–14).

Point 12. of some tables the lines are not straight and it becomes difficult to read and compare data correctly. 

Response 12: Thank you for your comments. In order to make data to be better presented, we revised and reorganized (with lines) all the tables based on reviewer’s suggestion.

Point 13. line 318: The order is changed; please mention AEW first for clarity. It is nice when this is consistent along the whole paper.

Response 13: Thank you for your comments. We appreciate this reviewer’s comments. In order to make data to be better presented, we reorganized and mention AEW intervention first for clarity based on reviewer’s suggestion as follows (Please see line 304–305 on page 9).

Point 14. It is not easy to read line 335-365 and Table 5 and 6. Remove duplicate information. Is description in results section possible with the data in Table 5 and 6?

Response 14: Thank you for your comments. In order to make data to be better presented, we reorganized the description of table 5 and 6 based on reviewer’s suggestion as follows (Please see line 321–344 on page 11).

“The intervention effects on biomarkers and QoL after the 8-week intervention are shown in Table 5 and 6. There were significant within-time-induced differences in AGEs, MDA, the NLR, and WBCs before and after the 8-week intervention, but no significant within-time-induced differences in AOPPs, blood glucose, and QoL parameters before and after the 8 weeks. Moreover, there was no differences in any biomarkers and QoL parameters between each intervention group and the control group at the baseline. After full adjustment, the significance of the interaction group x time analysis for all biomarkers revealed that participants in the AEW group exhibited significant reductions in AOPPs (ß=-0.185; 95% CI=-0.302~-0.068), AGEs (ß=-0.298; 95% CI=-0.433~-0.163), MDA (ß=-0.566; 95% CI=-0.691~-0.440), blood glucose (ß=-54.583; 95% CI=-96.085~-13.080), the NLR (ß=-1.666; 95% CI=-1.771~-1.560), and WBCs (ß=-2.406; 95% CI=-2.687~-2.125), but increased in SF36-PCS (ß=14.034; 95% CI=10.727~17.342) and SF36-total QoL scores (ß=6.828; 95% CI=4.099~9.557) after the 8-week intervention compared to the control group. Besides, participants in the regular walking group had significant declines in AOPPs (ß=-0.184; 95% CI=-0.313~-0.055), AGEs (ß=-0.304; 95% CI=-0.445~-0.163), MDA (ß=-0.579; 95% CI=-0.727~-0.431), blood glucose (ß=-49.680; 95% CI=-92.196~-13.080), the NLR (ß=-1.623; 95% CI=-1.713~-1.533), and WBCs (ß=-2.020; 95% CI=-2.289~-1.752),  but increased in SF36-PCS (ß=18.017; 95% CI=14.170~21.865), SF36-MCS (ß=13.360; 95% CI=7.522~19.198), and SF36-total QoL scores (ß=15.689; 95% CI=11.964~19.414) after the 8-week intervention compared to the control group. Moreover, compared to the control group, participants in the combined AEW with regular walking group also had significant reductions in AOPPs (ß=-0.264; 95% CI=-0.454~0.074), AGEs (ß=-0.364; 95% CI=-0.546~-0.182), MDA (ß=-0.716; 95% CI=-0.909~-0.524), blood glucose (ß=-57.223; 95% CI=-99.111~-7.163), the NLR (ß=-1.798; 95% CI=-1.897~-1.897), and WBCs (ß=-2.833; 95% CI=-3.119~-2.547), but higher SF36-PCS (ß=24.483; 95% CI=20.881~28.805), SF36-MCS (ß=25.649; 95% CI=20.310~30.988), and SF36-total QoL scores (ß=25.068; 95% CI=21.961~28.175) after the 8-week intervention (Table 5 and 6)”.

Point 15. Is it possible that the hydrogen has still special properties after consumption? Is this not lost in the stomach in the interaction with food and drinks in the stomach?

Response 15: Thank you for your comments. We appreciate this reviewer’s. Let us clarify and revise this point.  

“AEW has extremely high dissolved molecular hydrogen properties [18]. Hydrogen can rapidly diffuse across cell membranes, and this may provide powerful protection against oxidative stress through its ability to bind with hydroxyl radicals [50] (Please see line 402–450 on page 15).

Also, in this revised manuscripts, we added above point of “We did not measure the pharmacokinetics of AEW in the gastrointestinal of the subjects. Therefore, future research should investigate the molecular mechanisms of hydrogen at physiologically such as gastrointestinal” into our limitation (Please see line 447–449 on page 16).

Point 16. The Discussion is a bit messy with respect to information; it jumps from topic to topic, not easy to follow. Combine parts that discuss the same topic (e.g. antioxidants in line 411-415 and in line 444-447).

Response 16: Thank you for your valuable comments. We have combined the same topic together (Please see line 397–400 on page 15).

Point 17. Ln 429-437 all kind of topics; try to focus for a better/more effective discussion of the results.  

Response 17: Thank you for your valuable comments. We have removed “As we know, glycation occurs when excessive simple sugars such as fructose or glucose react non-enzymatically with amino groups on proteins, lipids, and nucleic acids [51]” based on reviewer suggestion.

Point 18. Ln 458: increased blood glucose? Don't you mean decreased?

Response 18: Thank you for your valuable comments. Let we clarify and revise this point to make it more clearer and precise. We have revised the word “increased” to “decreased” (Please see line 438 on page 16).

Point 19. The results of the SF-36 are not functionally explained; felt subjects better? What did subjects report? Did this affect their compliance? The relation with the inflammation markers is not so logic, so please elaborate on that.

Response 19: Thank you for your valuable comments. We appreciate this reviewer’s comments. Let us clarify and revise this point. 

“Our findings also support our secondary hypothesis that the combined AEW and regular walking group was significantly associated with a higher QoL as assessed by SF-36 scores, including PCS, MCS, and total QoL scores, which has good performance of physical and mental condition” (Please see line 434–437 on page 16).

Also, We did not measure the effect of SF-36 QoL on their medication compliance. However, “our study revealed that overall compliance patients to the AEW and regular walking intervention was optimal” (Please see line 279–280 on page 7).

We have added some information about the relation of QoL with inflammation markers as follow “Moreover, lower levels of inflammation such as NLR and WBCs were positively associated with higher QoL [5]. Similarly, amelioration of chronic inflammation is a potentially novel biobehavioral therapeutic strategy for improving QoL [10]” (Please see line 95–100 on page 3).

Point 20. Were physical effects found? Did you measure? Can you refer to that paper or mention main results? 

Response 20: Thank you for your valuable comments. In our results (Table 4 and 6) showed that synergistic effects of AEW and regular walking positively increase SF36-physical component score (PCS).

Point 21. In the conclusion you come up with the relevance of the interventions to be advised by health care professionals. However, in the paper no discussion is present about the community set-up (why is this so special) and are nurses able to do provide this information (time, interaction, education) to patients.

Response 21: Thank you for your valuable suggestion. We have removed the conclusion sentence “Our study indicated that health professionals educators have important role to promote nursing care strategies such as increase physical activity and frequently consume antioxidant-based drinking water to control FBG, inflammation markers, stress oxidative and improve quality of life”.

We have added some information about the advantage of community set-up as follow “Community-based health program are practical, relatively low-resource, often requiring lifestyle-changing educational programs. Previous study suggested that community-based RCT can diagnose as well as monitor how well diabetes is controlled and improve community health habits, which ultimately reinforce the high priority given to research translation into new practices (Please see line 360–364 on page 15).

Point 22. Please read the text once more carefully because some typing errors and not completely correct sentences are present in the text. Some plural forms are mentioned when a single form is needed etc.

Response 22: Thank you for your valuable suggestion. This manuscripts was edited by Taipei Medical University Academic Editing.

Reviewer 2 Report

The manuscript submitted for publication by Rias et al., titled: “Synergistic Effects of Regular Walking and AlkalineElectrolyzed Water on Decreasing Inflammation andOxidative Stress, and Increasing Quality of Life inIndividuals with Type 2 Diabetes: A CommunityBased Randomized Controlled Trial” is attempting to evaluate the effect of walking and alkaline electrolyzed water on inflammation and oxidative stress as well as quality of life on T2DM patients.

Below are the reviewer’s comments:

  1. The authors did not measure critical parameters of T2DM management such as fasting plasma glucose and HbA1c. Thus it is unclear as to if and to what extent the treatments help with T2DM management and progression.
  2. In the abstract in the last sentence the authors imply that walking and consumption of AEW is a “feasible treatment” for T2DM. This is a very strong statement and was not actually something that the experimental design of this work measured and should thus be removed.
  3. In the first sentence of introduction the authors state that there is absolute impairment of insulin secretion in T2DM. This is inaccurate and fundamentally wrong as a statement. In fact in the early stages of T2DM due to insulin resistance there is an overproduction and secretion of insulin albeit with progressively lower efficiency in terms of glucose clearance as insulin resistance establishes and intensifies. At the very late states of T2DM we may have insulin secretion reduction to very low levels due to pancreatic beta cell death due to exhaustion but for the majority of a diabetics life insulin is actually produced to a larger or smaller degree by their pancreas.
  4. The issue with oxidative stress/inflammation and T2DM is to a large extent a chicken-egg problem. It is clear that oxidative stress and inflammation are risk factors for T2DM but as T2DM onsets, the impaired glucose clearance induces oxidative phenomena and inflammation. This is something that needs to be discussed both in the discussion and in the introduction to provide a more accurate context.
  5. While the authors took significant care in selecting the participants in a systematic way illustrating inclusion/exclusion criteria, power calculations for sample size determination etc, it is unclear how they were able to conduct ceteris paribus comparisons among groups. For instance how did they know that the control group had similar dietary habits with the non-control/treatment groups? Also what was the status and use of medication (diabetes related medication) and how was that controlled as a confounding factor? Diabetes management depends significantly on dietary management and medication additional to physical exercise.
  6. Smoking is a very strong factor in terms of ROS and induction of inflammation. How were results normalized especially when the level of smoking is difficult to assess?
  7. Flow and grammar/syntax typos need to be improved.

Author Response

RE: antioxidants-929041-Version 1

Response to Reviewer 2 Comments

The manuscript submitted for publication by Rias et al., titled: “Synergistic Effects of Regular Walking and Alkaline Electrolyzed Water on Decreasing Inflammation and Oxidative Stress, and Increasing Quality of Life in Individuals with Type 2 Diabetes: A Community Based Randomized Controlled Trial” is attempting to evaluate the effect of walking and alkaline electrolyzed water on inflammation and oxidative stress as well as quality of life on T2DM patients.

Below are the reviewer’s comments:

Point 1. The authors did not measure critical parameters of T2DM management such as fasting plasma glucose and HbA1c. Thus it is unclear as to if and to what extent the treatments help with T2DM management and progression.

Response 1: Thank you for your valuable suggestion. We have measured the effects of AEW and regular walking on fasting blood glucose (FBG). However, present study was not able to measure HbA1c due to higher cost and longer time duration to detect (> 3 months). We have added this in our limitation (Please see line 450–451 on page 16).

Point 2. In the abstract in the last sentence the authors imply that walking and consumption of AEW is a “feasible treatment” for T2DM. This is a very strong statement and was not actually something that the experimental design of this work measured and should thus be removed.

Response 2: Thank you for your comments. We appreciate this reviewer’s comment. We have revised the word “feasible” to “advisable” (Please see line 37 on page 1).

Point 3. In the first sentence of introduction the authors state that there is absolute impairment of insulin secretion in T2DM. This is inaccurate and fundamentally wrong as a statement. In fact in the early stages of T2DM due to insulin resistance there is an overproduction and secretion of insulin albeit with progressively lower efficiency in terms of glucose clearance as insulin resistance establishes and intensifies. At the very late states of T2DM we may have insulin secretion reduction to very low levels due to pancreatic beta cell death due to exhaustion but for the majority of a diabetics life insulin is actually produced to a larger or smaller degree by their pancreas.

Response 3: Thank you for your valuable suggestion. We appreciate this reviewer’s comment.

We have revised the definition of T2DM as follows “T2DM is characterized by insulin resistance where there is an overproduction and secretion of insulin in the early stage which lead to reduce insulin secretion and pancreatic beta-cell death at late states” (Please see line 43–44 on page 2).

Point 4. The issue with oxidative stress/inflammation and T2DM is to a large extent a chicken-egg problem. It is clear that oxidative stress and inflammation are risk factors for T2DM but as T2DM onsets, the impaired glucose clearance induces oxidative phenomena and inflammation. This is something that needs to be discussed both in the discussion and in the introduction to provide a more accurate context.

Response 4: Thank you for your valuable suggestion. We appreciate this reviewer’s comment. In this revised manuscripts, we added above point of “It has been reported that oxidative stress and inflammation are the risk factors for developing T2DM and major cause of the complications and mortality among patients with T2DM” (Please see line 44–46 on page 2).

Point 5. While the authors took significant care in selecting the participants in a systematic way illustrating inclusion/exclusion criteria, power calculations for sample size determination etc, it is unclear how they were able to conduct ceteris paribus comparisons among groups. For instance how did they know that the control group had similar dietary habits with the non-control/treatment groups? Also what was the status and use of medication (diabetes related medication) and how was that controlled as a confounding factor? Diabetes management depends significantly on dietary management and medication additional to physical exercise.

Response 5: Thank you for your valuable suggestion. We appreciate this reviewer’s comment.

In this revised manuscripts, we added a description to make a clear inclusion/exclusion criteria that all patients were on stable oral hypoglycemic agents (Metformin and/or Glibenclamide) (Please see line 145, 146 on page 4).

“Moreover, we excluded patients who were on insulin injection. All the medication were distributed by the physician and no changes were allowed” (Please see line 149–152 on page 4).  

We also adjusted dietary habit, including carbohydrate, protein, fat, fast-food, and fiber consumption between interventional group and control group for controlling confounder of dietary habit (please see line 272-274 on page 7). Also, we have added in our limitation that the present study did not give dietary management, which may influence our findings (Please see line 451 on page 16).

Point 6. Smoking is a very strong factor in terms of ROS and induction of inflammation. How were results normalized especially when the level of smoking is difficult to assess?

Response 6: Thank you for your valuable suggestion. We appreciate this reviewer’s comment. In our study, the smoking status among four groups were not significant different (p=0.961). Thus, we did not adjust the results with the smoking status (Please see table 1 on page 8; smoking status).

Point 7. Flow and grammar/syntax typos need to be improved.

Response 7: Thank you for your valuable suggestion. This manuscripts was edited by Taipei Medical University Academic Editing.

Round 2

Reviewer 2 Report

The manuscript in its revised form is acceptable for publication. Some minor typos and grammar improvements would improve the flow and outlook further.